# The Xenon Road to Direct Detection of Dark Matter at LNGS: The XENON Project

Pietro Di Gangi 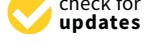 on behalf of the XENON Collaboration

Department of Physics and Astronomy, University of Bologna and INFN-Bologna, 40126 Bologna, Italy; digangi@bo.infn.it

**Abstract:** Dark matter is a milestone in the understanding of the Universe and a portal to the discovery of new physics beyond the Standard Model of particles. The direct search for dark matter has become one of the most active fields of experimental physics in the last few decades. Liquid Xenon (LXe) detectors demonstrated the highest sensitivities to the main dark matter candidates (Weakly Interactive Massive Particles, WIMP). The experiments of the XENON project, located in the underground INFN Laboratori Nazionali del Gran Sasso (LNGS) in Italy, are leading the field thanks to the dual-phase LXe time projection chamber (TPC) technology. Since the first prototype XENON10 built in 2005, each detector of the XENON project achieved the highest sensitivity to WIMP dark matter. XENON increased the LXe target mass by nearly a factor 400, up to the 5.9 t of the current XENONnT detector installed at LNGS in 2020. Thanks to an unprecedentedly low background level, XENON1T (predecessor of XENONnT) set the world best limits on WIMP dark matter to date, for an overall boost of more than 3 orders of magnitude to the experimental sensitivity since the XENON project started. In this work, we review the principles of direct dark matter detection with LXe TPCs, the detectors of the XENON project, the challenges posed by background mitigation to ultra-low levels, and the main results achieved by the XENON project in the search for dark matter.

**Keywords:** dark matter; direct detection; xenon; tpc

## 1. Introduction

Ordinary matter described by the Standard Model (SM) of particles contributes to less than 20% of the total mass budget of the Universe [1]. The remaining, dominant fraction consists of invisible dark matter (DM), whose nature is one of the most attractive and fundamental questions in physics nowadays. Compelling gravitational evidence supporting the DM existence has been collected in the last few decades at the galactic and galactic clusters scale [2]. While the postulation of some new elementary particle seems the most simple and natural solution to the DM enigma, modifications of the theory of gravity that would not require the presence of additional matter have been proposed to solve the observed gravitational anomalies [3]. Nonetheless, the strongest argument in favour of massive DM particles probably arises from the cosmological scale, where the successful ΛCDM model requires particle DM [4]. Moreover, the structure formation in the early Universe seems to be irreproducible without DM [5].

The DM quest is a golden channel for the search of new physics beyond the Standard Model (BSM) as no SM particles are viable DM candidates. A plethora of BSM particles and models have been proposed, as most of the basic DM properties remain obscure and the allowed mass range can span over fifty decades. The class of weakly interacting massive particles (WIMPs) [6] stands out as the most appealing proposal. WIMPs appear in a large variety of BSM models and with typical mass ($\sim$GeV/c$^2$ ÷ TeV/c$^2$) and annihilation cross-section of the weak scale, they are able to reproduce the observed relic DM density [7]. The naturalness of the WIMP hypothesis, also known as the "WIMP miracle", attracted the majority of experimental efforts in the last few years. Widely discussed candidates for light

DM are also axions [8] and, more generically, pseudoscalar axion-like particles (ALPs). For an extensive discussion on DM candidates, see Ref. [9]. The observation of a DM particle would clearly mark a historical breakthrough in the knowledge of fundamental physics and would open a new window for the exploration of the late and early Universe.

For nearly three decades, the experimental DM search programs have been continuously growing and intensifying their activity with impressive improvements of detection sensitivity. The global run for DM detection follows three complementary approaches: direct detection (DD) of DM particles scattering off target atoms, indirect detection (ID) of excesses over the astrophysical background due to DM annihilation, decay or conversion into SM particles [10], and DM production in high energy collisions with particle accelerators [11]. Among DD strategies using different technologies, ultra-low background time projection chambers (TPCs) with noble liquid target medium has achieved the strongest sensitivity to WIMP interactions. In particular, liquid Xenon (LXe) TPCs are leading the field over the last decade. A general discussion on the global DD program, its evolution, the current status, and prospects can be found in Refs. [12–14].

The LXe TPC technology is the foundation of the XENON project, which operated its first detector in 2005: the XENON10 prototype, with $\mathcal{O}(10)$ kg LXe target mass, installed at the INFN Laboratori Nazionali del Gran Sasso (LNGS). Since then, both XENON and LNGS have been the front-runners in the direct DM search, thanks to progressively improved detectors and by hosting numerous successful DM experiments, respectively. The legacy of XENON detectors continued with XENON100 ($\mathcal{O}(100)$ kg LXe mass) in 2008. With XENON1T, the XENON Collaboration opened the era of ton-scale DM detectors in 2016 and boosted the sensitivity of DD searches to currently unmatched levels. Further improvements are expected with the upgrade XENONnT installed in 2020 at LNGS and its first science run planned in 2021. The XENON Collaboration currently includes about 180 scientists from 27 institutions in 11 countries. The LXe community also features two other experiments with comparable sensitivity to XENONnT: LZ [15], located at the Sanford Underground Research Facility (SURF) in USA, and PandaX-4T [16], in the China JinPing Underground Laboratory (CJPL). The future generation of LXe TPCs will be represented by the proposed DARWIN observatory [17], a detector with 40 t LXe active mass aiming at the ultimate WIMP search with such technology.

The principles of the DD approach and the phenomenology of DM interactions are discussed in Section 2, specifically in the context of the LXe TPC technology (described in Section 3). The detectors of the XENON program are presented in Section 4. The background sources and the mitigation strategies adopted by the XENON project to fulfill the challenging requirements for the search of ultra-rare events are summarized in Section 5. Finally, the main results produced by the XENON experiments and the projections for XENONnT are discussed in Section 6.

## 2. Direct Detection of Particle Dark Matter with LXe Detectors

### 2.1. Direct Detection Principles

The idea that WIMPs could be detected through their collision with target nuclei in terrestrial detectors was brought up by Goodman and Witten [18] in 1985. This followed an earlier suggestion by Drukier and Stodolsky [19] to detect solar and reactor neutrinos via elastic neutral-current scattering off nuclei. Interestingly, such neutrino interaction is the ultimate background source for non-directional direct WIMP searches (as discussed in Section 5). As a consequence of DM nuclear scattering, the recoiling nucleus deposits energy in the medium through three processes: excitation of recoiling nuclei, which emit scintillation light via de-excitation, atomic ionization, providing detectable free charges, and heat production. The heat signal is measured from the phonons produced in crystals using cryogenic bolometers operated at $\mathcal{O}(\mathrm{mK})$ temperatures. Heat is also responsible for nucleation processes in experiments using superheated fluids. It is possible to observe two out of the three signals with different technologies: scintillating bolometers (phonon and scintillation light), bolometers with charge readout (phonon and charge), and dual-phase

noble-liquid TPCs (light and charge). The combined detection of two signals is crucial to disentangle DM signals from backgrounds to some extent, depending on the exploited technique (the case of LXe detectors is discussed in Section 3).

### 2.2. Dark Matter Signatures

Based on outstanding gravitational evidence [20,21], the presence of a galactic DM halo surrounding the Milky Way [22] is postulated. The Standard Halo Model (SHM) assumes that DM particles are trapped within the halo with velocities isotropically distributed according to the Maxwell–Boltzmann distribution. Typical parameter values describing the SHM are: local DM density $\rho = 0.3 \, \text{GeV/cm}^3$ [23], most probable DM velocity $v_0 = 220 \, \text{km/s}$ [24], and DM escape velocity (above which DM particles leave the potential well of the galaxy) $v_{esc} = 544 \, \text{km/s}$ [25] (in the galactic-rest frame). Recent recommendations on common parameter values to be adopted by any future DD experiment are summarized in [26], based also on modern observations [27].

The apparent DM velocity distribution $f(v)$ in the local frame also depends on the rotational motion of the solar system within the Milky Way with a velocity of $v_\odot \approx 232 \, \text{km/s}$ [28], on average. In principle, $f(v)$ has an annual modulation due to the Earth's orbital motion around the Sun at 30 km/s. Although the modulated part of the DM signal is only ~5%, dedicated DD experiments aim at observing an annually modulated DM signal above a fixed energy threshold [29]. In addition, the Earth's daily rotation can be exploited to search for diurnal modulation [30]. The apparent DM direction would constantly change during the day, while backgrounds are expected to be uniformly distributed or originate from the Sun's direction (for example, solar neutrinos). Background disentanglement can thus be achieved with directional DD experiments [31]. However, the measure of the DM track direction is experimentally challenging as the expected track length would be very small (~1 mm in gas and ~0.1 µm in solids) and thus difficult to reconstruct [32].

#### 2.2.1. DM–Nucleus Elastic Scattering

The primary physics channel for DD experiments is the elastic scattering of DM off target nuclei. In most scenarios, detectable signals are expected to arise from momentum transfer to a nucleus and the consequent nuclear recoil. The differential scattering rate (per unit recoil energy $E_R$ and target mass) can be written as [33]

$$\frac{dR}{dE_R} = \frac{\rho}{m_\chi m_N} \int_{v_{min}}^{\infty} v f(v) \frac{d\sigma}{dE_R} dv \tag{1}$$

where $m_\chi$ and $m_N$ are the masses of the DM particle and the target nucleus, respectively, and $\sigma$ is the DM–nucleus scattering cross section. DM is expected to be non-relativistic, hence the scattering kinematics can be described by Newtonian mechanics and the recoil energy is given by

$$E_R(v) = \frac{\mu_N^2 v^2}{m_N}(1 - \cos\theta) \tag{2}$$

where $\mu_N = \frac{m_\chi m_N}{m_\chi + m_N}$ is the DM–nucleus reduced mass and $\theta$ the scattering angle. Consequently, the minimum DM velocity $v_{min}$ to produce a nuclear recoil of a given energy $E_R$ is

$$v_{min} = \sqrt{\frac{m_N E_R}{2\mu_N^2}} \tag{3}$$

corresponding to head-on collisions, i.e., $\theta = \pi$. In the local frame, the cut-off of the $f(v)$ distribution, i.e., the maximum DM velocity, is $v_{esc} + v_\odot$. The maximum recoil energy as a function of DM velocity can be expressed as $E_{max}(v) = 2\mu_N^2 v^2/m_N$.

The scattering cross section can be described in terms of two components, corresponding to spin-independent (SI) and spin-dependent (SD) interactions [34]:

$$\frac{d\sigma}{dE_R}(v) = \frac{1}{E_{max}(v)}[\sigma_{SI}F_{SI}^2(E_R) + \sigma_{SD}F_{SD}^2(E_R)] \tag{4}$$

where $F_{SI}$ and $F_{SD}$, called form factors, carry the energy dependence and account for the loss of coherence due to quantum effects with increasing momentum transfers that suppresses the cross section. In fact, the DM particle scatters coherently with the entire nucleus at small momentum transfers, which is the case for typical <100 keV recoils of interest. The most generic description of DM–nucleus interactions relies on a plethora of relativistic and non-relativistic operators, in the context of the so-called *effective field theories* (EFTs) [34–36]. The SI and SD interactions are mainly represented by two specific non-relativistic EFT operators. The formulation of the form factors requires advanced nuclear physics calculations. A discussion on the parameterization of $F_{SI}$ and $F_{SD}$ can be found in [37,38], respectively.

The spin-independent cross section can be expressed in terms of the DM-nucleon cross section $\sigma_{nucleon}^{SI}$ as

$$\sigma_{SI} = \sigma_{nucleon}^{SI} \frac{\mu_N^2}{\mu_{nucleon}^2} A^2 \tag{5}$$

with $\mu_{nucleon}$ the DM-nucleon reduced mass and $A$ the mass number of the target nucleus. Such formulation holds when the coupling to protons and neutrons is assumed to be identical. Constraints on WIMP DM are usually reported in terms of $\sigma_{nucleon}^{SI}$ in order to compare between different target nuclei. The dependence on $A^2$ enhances the expected DM scattering rate in heavier nuclei, thus favoring Xenon ($A = 131.29$) over other lighter targets used in DD experiments (for example, Ge, Ar, and Si). SI interactions provide the largest expected rates and are thus the main search channel for WIMPs.

The spin-dependent component, instead, describes the coupling to unpaired nuclear spins and the cross section $\sigma_{SD}$ depends on the total nuclear spin of the target nucleus [39]. Results on the SD channel are usually reported for DM couplings to only protons or neutrons. Target nuclei with even number of nucleons can not effectively probe SD WIMP couplings. Natural Xenon contains two odd-nucleon isotopes ($^{129}$Xe and $^{131}$Xe, with an odd number of neutrons), thus enabling sensitive SD searches, in particular for WIMP coupling to neutrons.

### 2.2.2. Alternative DM Signatures

An alternative signature to standard WIMP elastic scattering are inelastic interactions where the nuclear recoil is accompanied by an electronic de-excitation of the target nucleus [40]. Analogous signature is expected by *inelastic dark matter* (iDM) models, where the DM particle itself gets excited [41]. The delayed coincidence between the two signals from inelastic scatterings can be used to reject backgrounds. However, the expected rates of inelastic interactions are significantly reduced compared to elastic scattering.

Recently, two processes have been proposed associated with ordinary DM–nucleus interaction: the *Migdal effect* [42,43] and the production of *bremsstrahlung* photons [44]. In the former case, atomic excitation and ionization would follow a nuclear recoil as the electron cloud is not dragged instantaneously together with the nucleus. The associated additional signal would help to detect low-energy DM scatterings below the experimental threshold. The bremsstrahlung effect similarly provides an additional signal, from photons in this case, but weaker than Migdal. Thanks to these effects, DD experiments can extend their sensitivity to lighter DM down to the MeV/c$^2$ regime, although neither has been proven experimentally yet.

Very light WIMPs with mass $\lesssim$ MeV/c$^2$ do not yield nuclear recoils of detectable energy in LXe detectors ($\gtrsim$keV). Nonetheless, they can scatter off atomic electrons and produce very small ionization signals corresponding to just a few or single electrons.

In some WIMP models, nuclear interactions are suppressed *leptophilic DM*) [45] and only WIMP-electron scattering can yield a detectable signal.

DM in the form of *axions* or *ALPs* are expected to induce a similar signal via the axio-electric effect [46,47], generically referred to as electronic recoil, contrary to the standard WIMP-induced nuclear recoils. The axion is absorbed and the atom is ionized. The electronic recoil energy corresponds to the sum of the rest mass of the axion and its kinetic energy, reduced by the electron binding energy.

## 3. Xenon-Based Dark Matter Detectors

### 3.1. Liquid Xenon Properties

The idea to use liquid Xenon as a medium for particle detection was first brought up in 1968 by the nobelist Luis W. Alvarez [48] and realized by building a proportional counter filled with LXe at Berkeley in 1971 [49]. Xenon is present in the atmosphere at the 400 ppb level by mass and pure Xenon gas is obtained by cryogenic separation of air. Worldwide pure Xenon production is about 70 t per year. Pure Xenon is therefore a rare and expensive consumable. Nowadays, more than 25 t of Xenon is used for DM searches across the world.

LXe-based experiments achieved the world-leading sensitivity to SI and SD WIMP-neutron interactions thanks to several advantages of Xenon as a target medium for DM detection. Xenon is a highly radiopure material, which is crucial for ultra-rare event searches. It has nine stable isotopes, while unstable ones are very short-lived, but $^{136}$Xe ($T_{1/2} = 2.2 \times 10^{21}$ y) and $^{124}$Xe ($T_{1/2} = 1.8 \times 10^{22}$ y). Both isotopes decay via second order weak processes though (double-beta and double-electron capture, respectively). They actually allow for new physics searches in the neutrino sector through their neutrinoless decay modes even without isotopic enrichment [50–52]. The two odd-spin isotopes $^{129}$Xe (spin 1/2) and $^{131}$Xe (spin 3/2), sensitive to the SD channel, are present with a large abundance of 26.4% and 21.2%, respectively. Thanks to the high atomic number ($Z = 54$) and a density of nearly 3 g/cm$^3$, LXe has excellent self-shielding properties against external radiation. In addition, this favors the construction of massive yet compact detectors. The relatively high triple point ($-112$ °C [53]) requires a not very challenging cryogenic system to keep the Xenon in liquid phase (typically at temperatures around $-95$ °C).

The deposited energy in LXe is shared among the production of electron–ion pairs, excited atoms, and subexcitation electrons (degrading into undetectable heat). Xenon is an excellent scintillator and energy depositions easily ionize it. The scintillation and ionization yields (i.e., photons and free electrons produced per unit energy) are some of the highest among other media. A fraction of the freed electrons recombine with ions forming additional excited atoms. Both direct excitation and electron–ion recombination produce excited Xenon dimeric molecules (excimers). The emission of scintillation light (luminescence) originates from excimers de-excitation to the ground state. The Xenon scintillation photons have a characteristic wavelength of 175 nm [54] to which Xenon itself is transparent. Excimers can be formed in a spin singlet or spin triplet state [55]. The luminescence process is therefore a combination of a fast and slow component, due to the singlet ($\sim$3 ns lifetime) and triplet ($\sim$24 ns) decays. The singlet to triplet ratio is dependent on the interaction type and energy [56,57] and could be used to discriminate between particles with the so-called pulse shape discrimination (PSD) technique. However, unlike in liquid Argon, the time separation is too small in LXe to achieve an effective PSD performance.

The number of excited atoms and electron–ion pairs is proportional to the total deposited energy and the respective amount of detectable light and charge quanta are highly anti-correlated [58]. Their fluctuations are also anti-correlated, hence the fluctuations of the combined light and charge signals are smaller than that of individual signals. Therefore, the capability to measure both light and charge signals improves the energy resolution of LXe detectors.

### 3.2. Dual-Phase LXe TPC Technology

The successful technology used by the XENON project to search for DM is based on dual-phase time projection chambers containing LXe with a small gap of gaseous Xenon (GXe) above it. A sketch of a dual-phase TPC and the signal production is shown in Figure 1. The cylindrical TPC is read by two arrays of light sensors, usually photomultiplier tubes (PMTs): one at the bottom immersed in LXe and one at the top facing the GXe region. The LXe sensitive volume is enclosed between two electrodes that establish an electric drift field: the cathode, negatively biased, at the bottom and the gate, grounded, at the top, right below the GXe volume. The drift field is kept uniform by means of thin copper rings properly biased and distributed along the vertical axis. A particle interaction in LXe produces a prompt scintillation signal, called $S1$, which is detected by both PMT arrays (in a larger fraction at the bottom due to internal reflection of scintillation photons at the liquid–gas interface). The ionization electrons that do not recombine are drifted towards the gaseous region at the top, where they are extracted by a strong electric field ($\mathcal{O}(10\,\text{kV/cm})$) applied by an anode electrode placed close to the gate, above the liquid–gas interface. Ionization electrons moving under a high electric field can acquire enough energy to excite atoms in GXe and produce light via proportional scintillation (electroluminescence) [59]. The related signal, called $S2$, is recorded by both PMT arrays. The $S2$ signal is delayed with respect to $S1$ by the electron drift time from the interaction site to the liquid–gas interface.

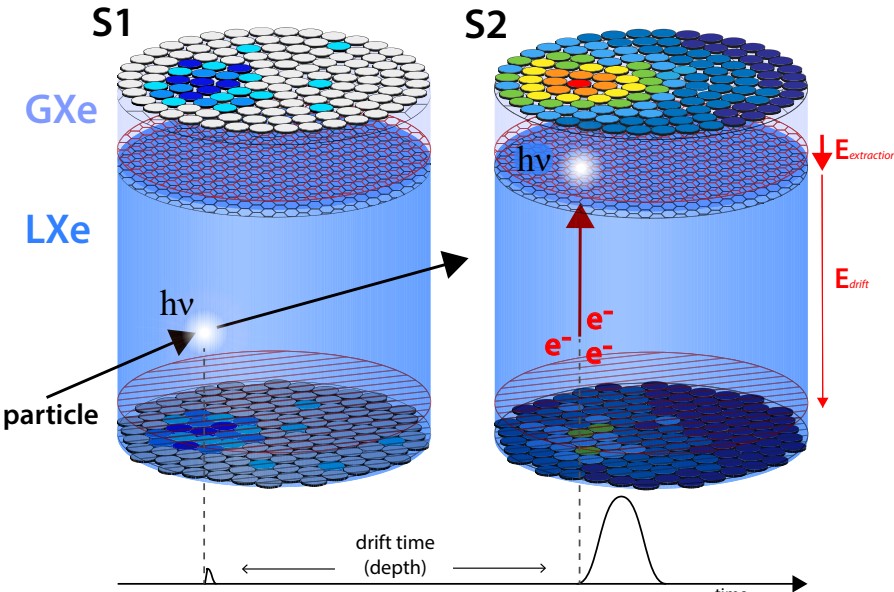

**Figure 1.** Schematic view of the working principle of a dual-phase TPC. The prompt scintillation signal ($S1$) is observed by both the top and bottom PMT arrays. Ionization electrons are drifted from the interaction vertex towards the LXe/GXe interface by means of a uniform electric field ($E_{drift}$) between the cathode (bottom red) and gate (right below the liquid–gas interface, black) electrodes. The $S2$ signal is formed via proportional scintillation triggered by electrons extracted in the gaseous region. The intense extraction electric field ($E_{extraction}$) is established between the gate electrode and the anode (top red). The top (bottom) screening electrode is also shown in black, right below (above) the PMT array. The localized pattern of the S2 signal in the top PMT array is used to reconstruct the interaction position in the $(x, y)$ plane. The time delay between $S1$ and $S2$ informs about the $z$-coordinate. The energy is reconstructed from the combination of both $S1$ and $S2$ signals.

This detection technique allows a three-dimensional reconstruction of the vertex position in the sensitive volume. The $(x, y)$ coordinates are determined from the hit pattern of the $S2$ signal in the top PMT array. The vertical position ($z$) is inferred based on the time difference between $S1$ and $S2$, as the electron drift time is proportional to the depth in the TPC. The 3D position reconstruction enables the selection of events restricted to the inner core of the LXe target, usually called fiducial volume (FV). Since the majority of background events are expected at the edges of the TPC, the outermost LXe volume is used as a shield to suppress the external backgrounds (see Figure 2).

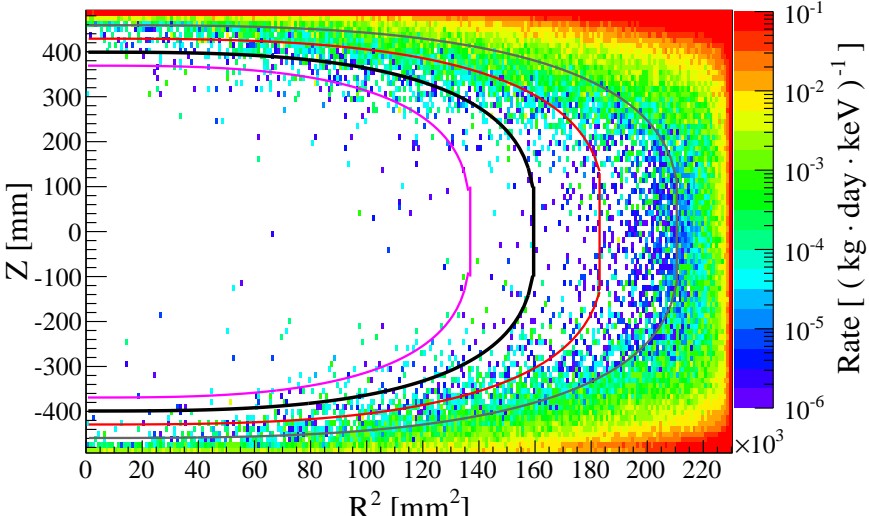

**Figure 2.** Simulation of background events originating from detector's materials radioactivity in XENON1T. The spatial distribution inside the active volume of the TPC is shown as a 2D histogram. The contours indicate progressively smaller super-ellipsoid inner fiducial volumes where the total background rate is largely suppressed with respect to the outermost layers of the TPC. The highly-efficient self-shielding extends also to external neutron sources. Image from Ref. [60].

The signal amplitude is corrected for position-dependent effects in the TPC. Both the position reconstruction itself and the detector response are usually characterized by injecting a $^{83m}$Kr calibration source that easily diffuses in the whole LXe volume. The $S1$ signal is corrected for the relative light collection efficiency (LCE) as

$$cS1 = \frac{S1}{\text{LCE}(x, y, z)} . \tag{6}$$

The LCE is smaller at the top due to internal reflection of photons at the liquid–gas interface and slightly decreasing towards the lateral edge of the TPC. The observed $S2$ amplitude is reduced by electron attachment to electronegative impurities in LXe (mostly $H_2O$ and $O_2$). The loss of charges exponentially depends on the time needed to drift the electrons up to the GXe gap, and it is parameterized by the so-called electron lifetime $\tau_e$. The Xenon is continuously circulated through a purification system where impurities are removed by hot getters. The LXe purity could in principle also impact the $S1$ signal via light absorption primarily on $H_2O$. However, such effect is much smaller and typically negligible even for large TPCs. The generation of the proportional scintillation is a highly localized process, confined between the LXe/GXe interface and the anode. Even small electrode warping or tilt results in large-scale inhomogeneities of $S2$s, especially in large TPCs. Therefore, in addition to the electron lifetime correction, the $S2$ signal is corrected also for the $(x, y)$-dependency, usually referred to as relative $S2$ LCE:

$$cS2 = S2 \times \frac{1}{e^{-t_{drift}(z)/\tau_e}} \times \mathrm{LCE}_{S2}(x,y) . \tag{7}$$

The detector performance in terms of light and charge signals detection is described by the primary scintillation gain $g1 = cS1/n_\gamma$ and the secondary scintillation gain $g2 = cS2/n_e$, respectively, where $n_\gamma$ and $n_e$ are the number of photons and electrons produced by an interaction. The energy is thus reconstructed from the $S1$ and $S2$ signals as

$$E = \left( \frac{cS1}{g1} + \frac{cS2}{g2} \right) \times W \tag{8}$$

where $W = 13.7 \pm 0.2$ eV [61] is the average energy required to produce a detectable quantum in LXe (either a photon or an electron). The energy scale is calibrated using sources of known mono-energetic lines that can be deployed outside the detector (e.g., $^{232}$Th), injected into the LXe target ($^{83m}$Kr, $^{37}$Ar) or activated during neutron calibrations ($^{129m}$Xe, $^{131m}$Xe). In highly sensitive experiments, even the low residual radioactivity of detector materials can be exploited ($^{40}$K, $^{60}$Co). Each mono-energetic line is selected with elliptical fits in the ($cS1$, $cS2$) space and light and charge yields are derived as the ratio of the observed signal amplitudes and the known energy of absorbed gammas. The detector-specific $g1$ and $g2$ parameters are estimated by fitting the anti-correlation of the measured light and charge yields from sources of different energy. Excellent linearity is typically observed from tens of keV up to few MeV [62,63]. The energy resolution is evaluated from the broadening of $cS1$ and $cS2$ signals. Mono-energetic sources are also used to monitor the time stability of the detector response during the multi-year operation timeline required for DM searches.

A key feature of Xe-based dual-phase TPCs is the highly efficient discrimination of nuclear recoils (NRs) from electronic recoils (ERs). The former are induced by WIMP scatters (and neutrons as well), while ERs include $\beta$ and $\gamma$ interactions originated from radioactive backgrounds. The electron–ion recombination process determines the ratio between $S2$ and $S1$ signal, as it subtracts electrons to the charge signal and enhances the scintillation light production. The extent of recombination depends on the particle type as tracks with high density of free charges lead to larger recombination rates. Highly ionizing particles like $\alpha$ deposit most of their energy in a dense electron–ion pairs core with a fraction of collected charges $\lesssim 10\%$, while almost all charges are collected for electron and $\gamma$ interactions (ERs). NRs produce dense tracks due to the high atomic number of the recoiling Xe nuclei, thus leading to a much larger recombination fraction than ERs. Therefore, the $S2/S1$ ratio of NRs is much smaller compared to ERs. As an example, the separation between the ER and NR populations is illustrated in Figure 3.

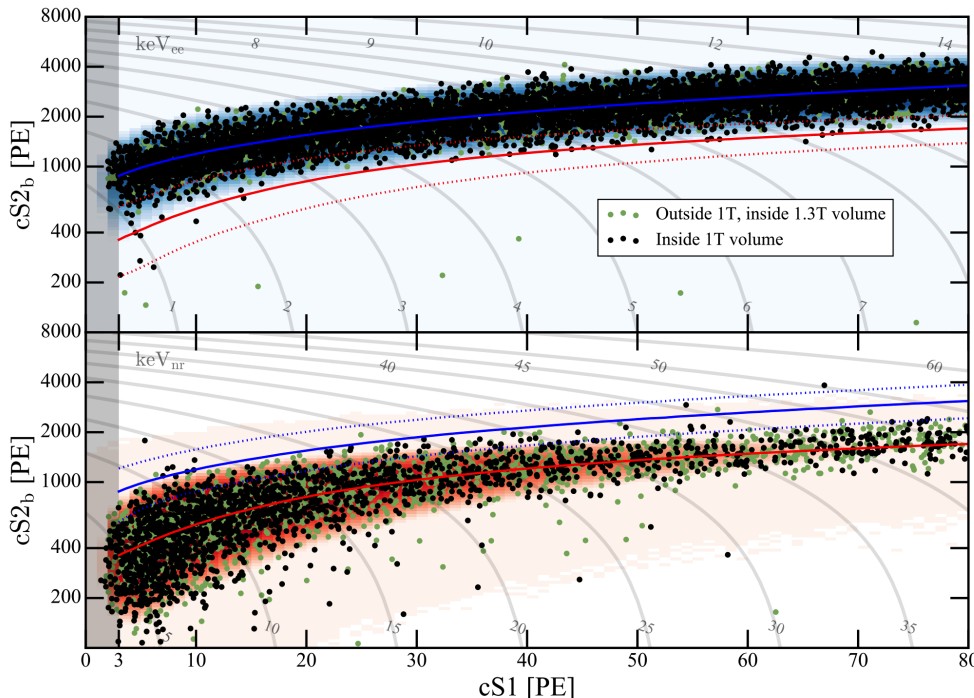

**Figure 3.** Separation between ER (top panel) and NR (bottom) populations as observed in the XENON1T TPC. ER data points are collected with a $^{220}$Rn calibration source, while NR events are produced by a Deuterium–Deuterium fusion neutron generator. Data points are shown within an inner 1 t core volume (black) and outside within a 1.3 t volume (green). The models extracted from the calibration data for ER (blue) and a 200 GeV/c$^2$ WIMP (red) are shaded and additionally drawn as 10%–50%–90% (dotted-solid-dotted) contour lines. The energy axes for ER (top) and NR (bottom) are drawn as grey contours. The ER discrimination power is usually evaluated in the (cS2/cS1 vs. cS1) space. The ratio between S2 and S1 is typically referred to as the discrimination parameter. Image from Ref. [64].

In the so-called discrimination space (cS2/cS1 vs cS1), >99% ER rejection power is typically achieved at 50% NR acceptance [65].

Overall, the scintillation yield for NRs is reduced compared to ERs due to quenching effects [66], as most of the energy is converted into atomic motion via exciton collisions and finally into heat. As a consequence, the experimental energy threshold is higher for NRs. LXe TPCs are able to reach very low-energy thresholds of ∼1 keV (for ERs) and ∼4 keV (for NRs). This is crucial for a highly sensitive DM experiment, in particular for low-mass WIMPs. Using the S2 signal only, the threshold can be further reduced to search for ≲6 GeV/c$^2$ WIMPs at the expense of increased background [67].

## 4. Detectors of the XENON Project

One of the advantages of the TPC technology is the simple scalability to larger detectors. The XENON project started with the XENON10 detector in 2005, whose active target mass was 15 kg. Over the last 16 years, the LXe target mass was increased by a factor ∼400 through the XENON100 and XENON1T experiments, up to today's 5.9 t of XENONnT. Even more impressive is the strive to achieve ultra-low background levels that resulted in a suppression by more than 4 orders of magnitude from XENON10 to the current expectation for XENONnT. The evolution of the XENON TPCs in terms of size and background level is sketched in Figure 4. In the following, each detector of the XENON project is described with particular focus on the recent ton-scale XENON1T and XENONnT TPCs.

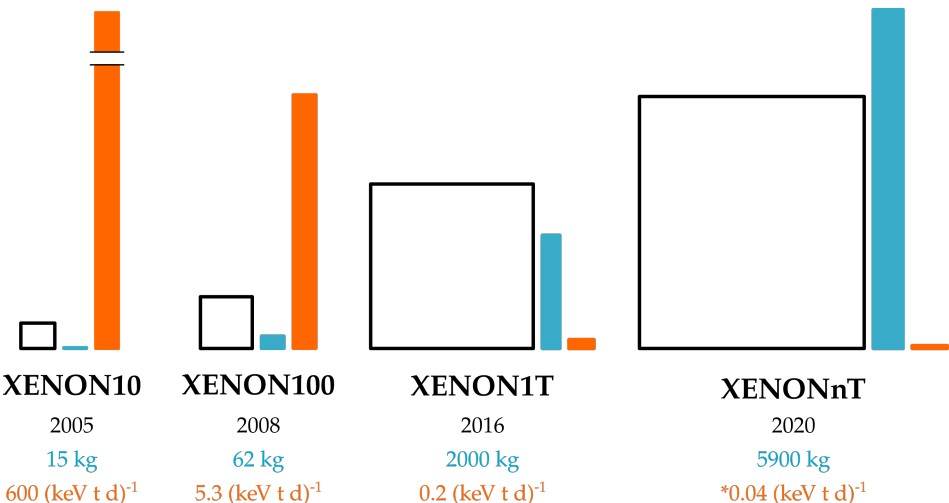

| XENON10 | XENON100 | XENON1T | XENONnT |
|---|---|---|---|
| 2005 | 2008 | 2016 | 2020 |
| 15 kg | 62 kg | 2000 kg | 5900 kg |
| 600 (keV t d)$^{-1}$ | 5.3 (keV t d)$^{-1}$ | 0.2 (keV t d)$^{-1}$ | *0.04 (keV t d)$^{-1}$ |

**Figure 4.** The detectors of the XENON dark matter search program. The dimensions of the cylindrical TPCs (diameter and height) are compared by the empty boxes with black contour (the relative scale and shape differences among the detectors are faithfully preserved): $20 \times 15$ cm (XENON10), $30.6 \times 30.5$ cm (XENON100), $96 \times 97$ cm (XENON1T), $132.8 \times 148.5$ cm (XENONnT). The year of construction is reported in black. The cyan bars and text show the active LXe mass (contained inside the TPC). The total amount of LXe used inside the cryostat is: 25 kg (XENON10), 161 kg (XENON100), 3200 kg (XENON1T), 8000 kg (XENONnT). In orange, the level of low-energy ER background ($\lesssim 20$ keV) is shown. The XENONnT rate (marked with *) is the experiment's goal according to the expected performance and purity.

### 4.1. XENON10

The XENON10 TPC [68] was built in 2005 and deployed underground in March 2006 in the interferometer tunnel of the LNGS underground laboratory. The goal was to demonstrate the achievable energy threshold and ER background rejection power [68]. The TPC active volume was defined by a polytetrafluorethylene (PTFE, Teflon) cylinder 15 cm tall and 20 cm wide. PTFE ensures high reflectivity for the LXe scintillation light [69] and electrical insulation. The photosensors used to detect the $S1$ and $S2$ signals were Hamamatsu R8520-06-Al 1"-square PMTs (Shizuoka, Japan) with a bialkali photocathode and quartz window, providing a $>20\%$ quantum efficiency at 175 nm wavelength. A total of 89 PMTs was employed, with 48 in the top and 41 in the bottom array. The TPC was enclosed in a vacuum stainless steel (SS) cryostat in order to maintain the LXe temperature at $-93\ ^\circ$C. The cooling power was provided by a pulse tube refrigerator [70] for a reliable and stable operation over several months. The total amount of LXe contained in the cryostat was 25 kg, including the external portion surrounding the TPC used as passive shield.

A cubic steel-framed structure, consisting of 20 cm high-density polyethylene (HDPE) inside 20 cm of low-radioactivity Lead, shielded the detector from external backgrounds.

### 4.2. XENON100

The XENON100 detector [71] was designed to increase the target mass and lower the background level by 1 and 2 orders of magnitude, respectively. XENON100 replaced XENON10 in the interferometer tunnel of LNGS in 2008 and ended data acquisition operations in 2016. The TPC, made of 24 PTFE panels, had a height of 30.5 cm and a diameter of 30.6 cm. In addition, 178 PMTs of the same model as XENON10 were selected for low radioactivity [72] and distributed in the top (98) and bottom (80) arrays of the TPC. The TPC was surrounded by a $\sim$4 cm layer of LXe, for a mass of 99 kg, observed by additional 64 PMTs. Such volume was operated as an active LXe veto in order to effectively reduce backgrounds by rejecting coincident events with the TPC [73]. In order to ensure the highest stability and precise control of the liquid level in the TPC, an SS diving bell enclosing the top PMT array of the TPC was designed. The diving bell system also allowed

for keeping the LXe level in the cryostat vessel arbitrarily high so that the LXe veto could also cover the volume above the TPC.

The passive shield used in XENON10 was improved by adding a layer of 5 cm-thick oxygen-free high thermal conductivity (OFHC) Copper and a ~20 cm layer of polyethylene or water on all sides to further suppress the neutron background. High purity boil-off Nitrogen was constantly flushed into the shield cavity in order to remove radioactive Radon that could penetrate the shield. The cryogenic system was moved away from the detector, outside the passive shield, establishing a remote cooling operation to avoid radioactive background originating from components of the cryogenic system itself.

*4.3. XENON1T*

The XENON1T detector [62], shown in Figure 5, became operational at LNGS in Spring 2016 and terminated its operations three years later to make room for the successor XENONnT. The detector was filled with total 3.2 t of LXe, of which 2.0 t in the active TPC volume. XENON1T was the first ton-scale LXe dark matter experiment ever built and the most sensitive DM detector to provide scientific results so far. The TPC was installed inside the cryostat in the center of a large (~10 × 10 m) water tank. The tank serves as a passive shield as well as a Cherenkov Muon Veto detector (see Section 5.2.2). A three-floor building erected in the Hall B of LNGS underground laboratory accommodates all ancillary systems. These include the systems to cool, purify, and store the Xenon gas, a cryogenic distillation column for Krypton removal (see Section 5.2.5), the data acquisition system as well as the control and monitoring systems for the entire experiment.

The cylindrical TPC of 97 cm height and 96 cm diameter was defined by 24 interlocking and light-tight PTFE panels, whose surfaces were treated with diamond tools in order to optimize the reflectivity for vacuum ultraviolet light. Due to the rather large linear thermal expansion coefficient of PTFE, its length is reduced by about 1.5% at the operation temperature of −96 °C. An interlocking design allowed the radial dimension to remain constant while the vertical length is reduced.

The electrodes to establish the drift and extraction fields and to screen the PMT arrays (gate, anode, and screening electrode on top, and cathode and screening electrode on bottom) were designed to maximize the $S1$ light collection. The top electrodes were hexagonal grids with 178 μm-thick SS wires (but the gate: 127 μm), while the bottom electrodes consisted of 216 μm-thick parallel Au plated SS wires. The cathode is negatively biased, the anode is positively charged, the gate electrode is kept at ground potential, and the screening electrodes can be biased to minimize the field in front of the PMT photocathodes. To ensure drift field homogeneity, the TPC was surrounded by 74 field shaping electrodes made of low-radioactivity OFHC Copper and connected by two redundant resistor chains.

An SS diving bell, which was directly pressurized by a controlled gas flow, was used to maintain a stable liquid–gas interface between the gate and anode electrodes. The height of the liquid level inside the bell was controlled via a vertically-adjustable gas-exhaust tube. Possible tilts of the TPC were measured by means of four custom-made parallel-plate-capacitive level-meters installed inside the diving bell, with a precision of ~30 μm.

A total of 248 3″ PMTs (Hamamatsu R11410-21) were radially installed in the top array (127 PMTs) to facilitate radial position reconstruction, and packed as tightly as possible in the bottom array (121 PMTs) to maximize scintillation light collection efficiency. They featured an average room temperature quantum efficiency of 34.5% at 175 nm [74], a high photoelectron collection efficiency of 90% and were designed to operate stably in gaseous and liquid Xenon at cryogenic temperature [75,76]. All installed PMTs were screened for their intrinsic radioactivity levels [54] and tested at room temperature and under gaseous Nitrogen atmosphere at −100 °C. The PMTs with the highest quantum efficiency were installed at the center of the bottom array to maximize the light collection efficiency. Both arrays consisted of a massive OFHC copper support plate with circular cut-outs for the PMTs. A single PTFE plate held the individual PMTs and a PTFE reflector plate covered

the area between the PMT windows. Custom-developed low pass filters installed on each high voltage and return line reduced the electronic noise to sub-dominant levels.

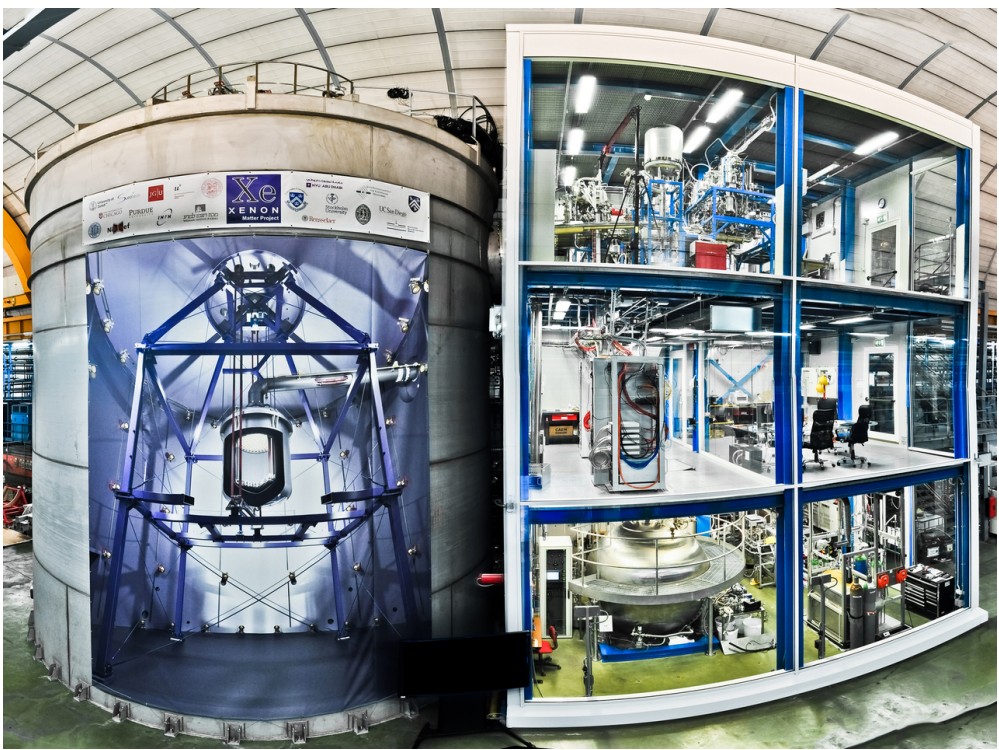

**Figure 5.** The XENON1T experiment in the Hall B of the LNGS underground laboratory. On the left, the 10 m-high water tank instrumented as an active water Cherenkov Muon Veto (see Section 5.2.2). The banner affixed to the side of the water tank illustrates the infrastructure enclosed inside, with the LXe TPC located at the center. On the right, the three-floor service building which hosts the ancillary systems: the cryogenic and GXe purification systems (top floor), the data acquisition and slow control equipment (middle floor), and the emergency LXe recovery and storage system (ground floor). The 5.5 m-high cryogenic Krypton distillation located on the ground floor extrudes into the middle floor. The XENONnT experiment reuses the XENON1T's infrastructure and systems, with the addition of a Neutron Veto detector (see Section 5.2.4), built around the TPC, a Radon distillation column, a LXe purification loop, and an expanded recovery and storage system.

The double-walled, cylindrical stainless steel cryostat was made of low-radioactivity material [77]. The inner vessel was 1.96 m high and 1.10 m in diameter. Its inner surface, in direct contact with the liquid xenon, was electro-polished in order to reduce the emanation of Radon. It was enclosed by an outer vessel of 2.49 m height and 1.62 m diameter. The connections to outside the water tank were made through a double-walled cryogenic pipe enclosing all the connections to the cryogenic system (cooling, purification, fast emergency recovery, and diving bell pressurization) and the cables for the PMTs and auxiliary sensors.

XENON1T followed the remote cooling concept successfully employed by XENON100. It allows for maintenance of the cryogenic system, which is installed far away from the TPC, while the detector is cold. The same system is kept and operated for the XENONnT experiment. The Xenon gas inside the cryostat is liquefied and kept at its operating temperature by means of two redundant pulse-tube refrigerators (PTRs), each providing ~250 W of cooling power. The Xenon pressure inside the cryostat is kept constant by controlling the temperature of the active PTR cold finger using resistive heaters. Xenon gas from the inner cryostat vessel streaming to the cryogenic system is liquefied, collected in a funnel and flows back to the cryostat vessel, driven by gravity, in a pipe that runs inside the cryogenic tube. Another pipe carries LXe out of the cryostat, evaporates it in a heat

exchanger, and feeds it to the purification system. The purified Xenon gas is liquefied in the same heat exchanger and flows back to the cryostat.

Electronegative impurities, such as water or oxygen, are constantly outgassing into the Xenon from all detector components. Therefore, the gas must be continuously purified to reduce the impurities to $<10^{-9}$ $O_2$-equivalent level (ppb). The purification loop consists of a gas transfer pump, a mass-flow controller, and a high-temperature rare-gas purifier (getter). The latter removes oxide, carbide, and nitride impurities by forming irreducible chemical bonds with the getter material (zirconium). The purification system is also used to inject calibration sources into the detector (e.g., $^{83m}$Kr and $^{220}$Rn), which are dissolved in the Xenon gas.

A Xenon storage system, called ReStoX, was developed to address the operational challenges posed by the first multi-ton LXe experiment: mainly fast TPC filling and Xe recovery. It consists of a vacuum-insulated stainless steel sphere with 2.1 m diameter. Its volume (4.95 m$^3$) and the wall thickness (28 mm) allow for storage of up to 7.6 t of Xenon as a liquid, as a gas, and even as a super-critical fluid (being capable of withstanding pressures up to 73 bar).

### 4.4. XENONnT

The upgraded XENONnT detector replaced XENON1T in 2020, when the installation of the new systems was successfully completed at LNGS despite the problems raised by the COVID-19 pandemic. XENONnT was planned as a fast upgrade built upon the already existing XENON1T infrastructures. The major improvements are related to a new larger TPC, containing three times more LXe than XENON1T, a newly developed Neutron Veto sub-detector (described in Section 5.2.4), an improved Xenon purification system, and a novel Radon distillation column (see Section 5.2.5).

A total of 8.4 t LXe mass fills an elongated SS cryostat placed at the center of the water tank. The TPC is enclosed by 24 interlocking PTFE reflective panels forming a quasi-cylinder of 1.49 m height and 1.33 m diameter. The active target LXe mass amounts to 5.9 t. The same XENON1T PMT model is used, and the total number of photosensors is almost doubled to 494. Both top (253) and bottom (241) PMTs are distributed in a compact hexagonal structure to maximize the light collection efficiency. Parallel SS wires stretched onto SS rings are used for all the electrodes. Two concentric OFHC copper rings ensure the drift field uniformity with 71 and 64 wires in the inner and outer sets, respectively. The diving bell method to control the liquid level in the TPC is maintained. An additional port in the bottom dome of the cryostat allows for extracting LXe from the TPC to be circulated in a newly developed Xenon purification system in liquid phase, in addition to the already existing GXe purification loop (which is operated with increased flow with respect to XENON1T, thus better performing).

## 5. Backgrounds and Mitigation Strategies

### 5.1. Background Sources

Two main categories of background can be distinguished: electronic recoils (discussed in Section 5.1.1), mainly due to gamma and beta particles from radioactive decays, and nuclear recoils (discussed in Section 5.1.2), dominated by neutron scattering. Interactions of $\alpha$ particles from sources diffused in the LXe target or on detector surfaces are incompatible with DM signatures as the deposited energy is too high. However, they can fall in the region of interest (ROI) if the energy is partially lost in insensitive detector regions (e.g., ionization electrons trapped in the TPC walls, leading to suppressed S2 signals). Background events can also originate from detector artifacts, accidental $S1$–$S2$ coincidences, or noise.

The ROI to look for WIMP signals is chosen such that the spectrum of even high mass WIMPs ($\mathcal{O}(\text{TeV}/c^2)$) is almost entirely contained. Given the different detector response to ERs and NRs, this translates into typical upper energy boundary of $\sim$15 keV and $\sim$50 keV, respectively. The WIMP-nucleon cross section is so low that at most one single scatter, if any, is expected in the entire target volume. Thus, only low energetic ERs and NRs that

are not accompanied by other interactions inside the TPC, contribute to the background. To illustrate the spectra and the relative importance of the different background sources described in the following sections, Figure 6 shows the residual ER and NR backgrounds expected in the XENONnT experiment.

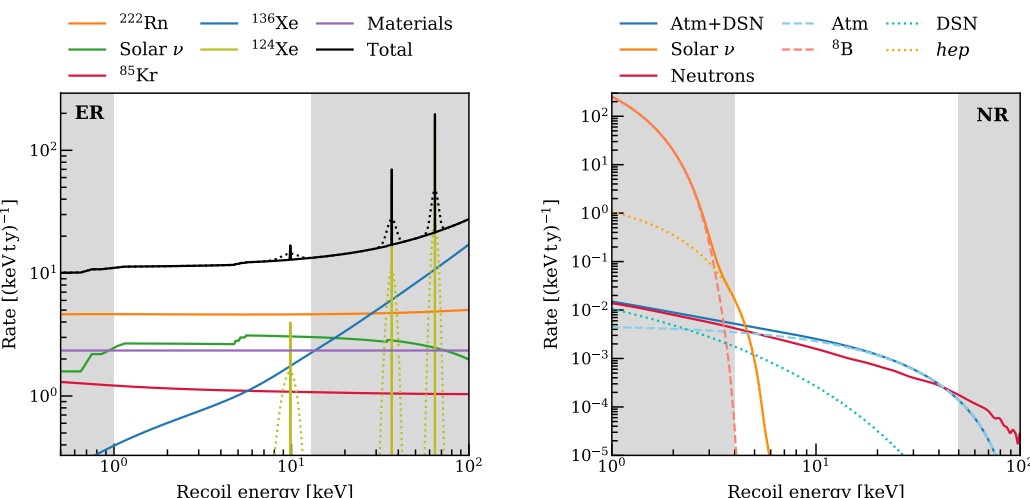

**Figure 6.** Energy spectra of ER (**left**) and NR (**right**) backgrounds expected in a core 4 t fiducial mass of the XENONnT detector. The white areas highlight the RoI for the standard SI WIMP search: (1, 13) and (4, 50) keV for ERs and NRs, respectively. The rate shown from $^{222}$Rn (orange line) assumes 1 μBq/kg activity concentration, while a 0.1 ppt concentration of natural Krypton in the LXe target is responsible for the $^{85}$Kr component (red). The dotted lines represent the Gaussian smearing of mono-energetic lines according to the energy resolution achieved in XENON1T [78]. The sources of NR background are distinguished into radiogenic neutrons (red), solar neutrinos (orange), and atmospheric (dashed blue) and diffuse supernova neutrinos (dotted blue). The neutron rate accounts for an expected 87% suppression thanks to the XENONnT's neutron veto system. Image from Ref. [79].

### 5.1.1. ER Backgrounds

Despite the good separation from the NR population, where the WIMP signal is expected, electronic recoils are the major source of background. High energy $\gamma$-rays ($\mathcal{O}(MeV)$) emitted by radioactive contaminants in detector construction materials surrounding the active LXe volume can produce low-energy ERs via Compton scattering. If they escape the detector after only one Compton scatter, without depositing their entire energy, external gammas can contribute to background. This source was dominant in XENON10 and XENON100, while the <10 cm penetration depth of MeV $\gamma$-rays in LXe [80] is small compared to the meter-scale XENON1T and XENONnT TPCs. The relevant contaminants in detector materials are long-lived natural radioisotopes ($^{238}$U, $^{232}$Th decay chains and $^{40}$K), cosmogenic ($^{60}$Co), and anthropogenic isotopes ($^{137}$Cs).

One source of electronic recoils below ~10 keV can be *X-ray photons* via photoelectric interaction. External sources of X-rays cannot lead to a background as they penetrate $\mathcal{O}(10~\mu m)$ in LXe. X-rays can be produced by radioactive contaminants in Xenon that undergo electron capture, such as $^{127}$Xe, which is cosmogenically activated. However, this background is negligible for Xenon inventories stored underground for several months or years like those used in the XENON experiments ($^{127}$Xe has a 36 day half-life). A cascade of X-rays and Auger electrons is emitted by the decay of $^{124}$Xe (isotope present with a 0.095% natural isotopic abundance) via double electron capture (DEC), a second order weak process directly observed for the first time by XENON1T [78], thanks to its unprecedented low background level. When the two electrons are captured from the L-shell (with a branching ratio of 1.7%), the expected ER signal has ~10 keV energy. The other electron capture lines produced by $^{124}$Xe, including the most probable one (at 64.3 keV, with 75%

branching ratio) that led to the DEC discovery, fall well above the ROI. $^{124}$Xe is now included the XENONnT background model, though subdominant, while it would have been completely negligible for the former XENON detectors. Remarkably, the measured $^{124}$Xe half-life of $1.4 \times 10^{22}$ y is the longest ever directly observed, demonstrating the excellent sensitivity of XENON1T (and XENONnT) to ultra-rare processes besides DM interactions [63], e.g., searches for new physics in the neutrino sector.

In the ton-scale detectors, the largest contributions to the ER background are due to $\beta$-emitter contaminants diffused in the LXe target. They produce the same signal as an electron recoil even though the moving electron is not produced by an actual collision. $\beta$-decays of heavy elements, such as the typical residual contaminants of ultra-pure Xenon, give rise to an almost flat spectrum below $\sim$10 keV as a consequence of their high kinematic endpoint energies. The hardest isotope to get rid of is $^{222}$Rn that can emanate from the detector materials, or diffuse through the seals. Due to its relatively long half-life (3.8 days), it can diffuse in the LXe volume almost homogeneously. Considering its daughters, down to the long-lived $^{210}$Pb, the contribution to low-energy ERs comes from the $\beta$-decay of $^{214}$Pb onto the ground state of $^{214}$Bi, with an end-point at 1023 keV, where no other radiation is emitted. If the decay occurs close to the borders of the active region, decays on higher energy levels are also potentially dangerous, since there is a finite probability that the accompanying $\gamma$ exits the detector with no energy deposition in LXe. This is responsible for a slight increase of the background rate from $^{222}$Rn towards the TPC edges. The only other dangerous $\beta$-emitter in the chain ($^{214}$Bi) can be easily removed by the time correlation with the $\alpha$-decay of its daughter ($^{214}$Po) which occurs with a half-life of 164 μs. The $^{220}$Rn isotope can also be emanated from detector materials, being a noble gas, but, due to its 56 s half-life, the probability to diffuse in the active LXe volume is much lower than $^{222}$Rn.

Commercially available Xenon is contaminated by Krypton with typical $^{nat}$Kr/Xe concentrations of $\mathcal{O}$(ppm). Natural Krypton contains traces of the radioactive isotope $^{85}$Kr, which is a product of nuclear fission, and it is released in atmosphere mainly by nuclear fuel reprocessing plants. Its relative isotopic abundance in Europe has been determined by low level counting to be $2 \times 10^{-11}$ [81]. $^{85}$Kr is an ER background source as it $\beta$-decays with half-life of 10.76 y and end-point energy of 687 keV.

The last intrinsic background source is $^{136}$Xe, which is contained in natural Xenon with 8.9% isotopic abundance. It decays via double-beta emission with Q-value 2458 keV and half-life of $2.17 \times 10^{21}$ y [82]. The induced background rate is subdominant with respect to $^{222}$Rn and $^{85}$Kr.

Solar neutrino scatterings off electrons are an external source of ER background, which cause single low-energy recoils uniformly distributed in the TPC (contrary to external gamma sources), given the weak cross section and the very long neutrino penetration depth. The dominant contribution comes from *pp* neutrinos, while $^{7}$Be, *pep*, and all other solar neutrino sources account for less than 10% of the total ER neutrino-induced background. This is an irreducible background for non-directional DD experiments. As the Xenon purification from $^{222}$Rn and $^{85}$Kr dramatically improved with respect to the early XENON experiments, solar neutrinos became relevant for XENON1T, and they are expected to be the second largest ER background contribution for XENONnT [79], comparable to $^{222}$Rn. Future multi-ton LXe TPCs could be able to measure with high precision the low-energy solar neutrino flux [50,83] with further suppression of the $^{222}$Rn contamination.

### 5.1.2. NR Backgrounds

Neutron interactions and coherent elastic neutrino-nucleus scatterings (CE$\nu$NS) can produce nuclear recoils which exactly mimic the WIMP signature. Neutrinos are expected to interact via CE$\nu$NS at most once in the TPC due to the weak cross section, while neutrons can scatter multiple times inside the fiducial volume, due to the $\sim$10 cm mean free path of MeV neutrons. Fast neutrons are more penetrating than $\gamma$-rays in LXe; therefore, shielding the inner LXe volume against neutrons is more difficult, and their probability to scatter only once in the active target mass is higher than $\gamma$s.

The presence of $^{238}$U, $^{235}$U and $^{232}$Th in the detector materials generates *radiogenic neutrons* in the MeV energy range through spontaneous fission (SF), mainly from $^{238}$U, and ($\alpha$, n) reactions induced by $\alpha$ particles emitted along the decay chains. For heavy nuclei, the high Coulomb barrier suppresses the ($\alpha$, n) process. Therefore, the neutron production is almost exclusively due to spontaneous fission in that case. The highest ($\alpha$, n) yields are from light materials, such as PTFE and the ceramic stems of PMTs. The neutron background is estimated mainly by Monte Carlo (MC) simulations. Its expectation is typically less than 1 event in DD experiments (depending on the fiducial mass choice and exposure). Hence, direct measurements of such background are very difficult. In the large XENON1T detector, the observation of multiple neutron scatters was used to set a constraint on the expected single scatter rate [84].

Muons penetrating the mountain rock and reaching the LNGS underground halls can produce *cosmogenic neutrons* with energies extending up to tens of GeV via spallation mainly in the cavern rock. Muon-induced neutrons are a potential source of NR background but can be reduced to a negligible level with respect to radiogenic neutrons by properly shielding the TPC (see Section 5.2.2) as they originate more far away from the detector.

Astrophysical and atmospheric neutrinos contribute to the NR background through CE$\nu$NS, a process predicted by the Standard Model difficult to detect but recently observed for the first time by the COHERENT experiment [85]. The most relevant contribution to the NR background comes from solar $^8$B and *hep* neutrinos. The induced recoil energy spectrum steeply falls down close to the average experimental NR energy threshold ($\sim$4 keV), with most of the expected event rate confined below the threshold. However, Poissonian fluctuations in the number of the generated scintillation photons allow for detecting part of the below-threshold events, finally yielding a non-negligible expected background rate in the ROI of $\sim$0.4 events per ton-year. Solar neutrino CE$\nu$NS impacts the sensitivity to WIMPs with masses of few GeV/$c^2$, as the induced recoil spectrum is almost identical to a 6 GeV/$c^2$ WIMP. In contrast, the more energetic atmospheric neutrinos and from diffuse supernovae (DSN) are relevant for heavier WIMP searches as their spectrum extends up to the upper ROI bound and beyond, analogously to the neutron spectrum. Their expected rate is about one order of magnitude lower than from solar neutrinos. The CE$\nu$NS background is irreducible in noble liquid TPCs and will represent the ultimate limitation to the WIMP sensitivity of DD experiments [86].

### 5.2. Background Mitigation

### 5.2.1. LNGS Underground Location

Direct searches for ultra-rare signals induced by DM particles in the galactic halo require an ultra-low background level. Therefore, they are conducted in deep underground laboratories. Past and present XENON experiments took place at LNGS [87], the largest underground research facility of the world for particle and astroparticle physics. The laboratory is located in central Italy under the Gran Sasso mountain and consists of three large experimental halls for a total area of 13,500 m$^2$. It provides an average rock overburden of 1400 m, corresponding to about 3800 meter water equivalent. The muon flux is reduced by a factor one million compared to the flux at the mountain surface, down to a value of 1.2 m$^{-2}$ h$^{-1}$ [88,89]. The limited content of Uranium and Thorium in the dolomite rocks of the Gran Sasso mountain ensures a thousand times less radiogenic neutron flux than on the surface, in the order of 1–10 m$^{-2}$ h$^{-1}$ [90] up to 5 MeV, depending on the neutron energy. Faster muon-induced neutrons ($>$10 MeV) are generated in the cavern rock with a flux of $\sim$2.7 $\times$ 10$^{-2}$ m$^{-2}$ h$^{-1}$ [91,92].

### 5.2.2. Muon Veto

In order to further suppress the cosmogenic neutron background, the XENON1T and XENONnT TPCs were placed inside a stainless steel cylindrical tank, 9.6 m in diameter and 10.2 m in height, filled with $\sim$700 t of deionized water which acts as a shield against external neutrons and gammas. Moreover, the tank is instrumented with 84 PMTs to

tag muon tracks and induced particle showers, through the detection of Cherenkov light produced in water. Such water Cherenkov sub-detector is called *Muon Veto* (MV) [93]. The 8" PMTs are distributed in five rings along the circumference of the water shield at different heights. To enhance the Cherenkov photon detection efficiency, the inner surface of the water tank is cladded with a foil >99% reflective at wavelengths between 400 nm and 700 nm [94].

Events observed in the TPC happening coincidently with a MV signal can then be recognized and rejected. The MV trigger requires an 8-fold PMT coincidence within 300 ns. Two classes of MV-tagged events can be distinguished: "muon" events, where the muon crosses the water tank, and "shower" events, in which the induced neutron comes inside together with the associated particle shower, but the muon does not enter the water tank. The tagging efficiencies of MV evaluated through MC simulations are 99.5% and 43% for "muon" and "shower" events, respectively. The combination of the water shield and the active MV tagging efficiency suppresses the cosmogenic neutron background by more than 2 orders of magnitude. The residual rate of $<0.01$ t$^{-1}$ y$^{-1}$ is negligible compared to the radiogenic neutron sources.

### 5.2.3. Radioassay of Detector Components

In order to reduce radiogenic ER and NR background sources, originating from radioisotopes in detector materials, all components of the TPC, the cryostat, and the surrounding structures are selected for a low content of radioactive isotopes with a thorough radioassay program. Measurements of the activity of relevant gamma-ray emitters (mainly $^{40}$K, $^{238}$U, and $^{232}$Th chains) are performed using low-background high-purity germanium spectrometers [95–97]. The most sensitive detectors are located in the LNGS underground facility and reach sensitivities down to the μBq/kg level. Complementary measurements employ high-resolution inductively coupled plasma-mass spectrometry (ICP-MS) at LNGS [98] to estimate the $^{238}$U and $^{232}$Th abundance with high accuracy. A detailed discussion on the radioassay for the XENON1T PMTs and all other detector components is given in Refs. [54,77], respectively.

Materials in contact with liquid or gaseous Xenon during standard operation are also selected for low $^{222}$Rn emanation rate. This includes most components of the TPC, the inner cryostat and its connection pipes, the cryogenic and Xenon purification systems. Although the $^{222}$Rn emanation rate is usually related to the $^{226}$Ra content of a material (that can be estimated by gamma-spectrometry), it must be measured independently since, in most cases, the emanation is dominated by surface impurities. The measurements are thus performed using dedicated $^{222}$Rn emanation facilities [99]. To remove radioactive isotopes from surfaces, all TPC components are cleaned after production according to specific procedures for each material type.

The TPC is assembled above ground at LNGS, inside a custom-designed cleanroom with a controlled particle concentration, using a movable transport frame. The double-bagged TPC (aluminized mylar) is then moved to the underground laboratory, and the installation is worked out inside a dedicated mobile soft-wall cleanroom.

### 5.2.4. Neutron Veto

Built for the XENONnT experiment, a Neutron Veto (NV) sub-detector is operated to reduce the previously dominant source of NR background: radiogenic neutrons. Similarly to the MV principle, XENONnT will be able to reject background events in coincidence with a neutron signal recorded in the NV. The largest fraction of the total radiogenic neutron background budget comes from the PMTs and the SS double-walled cryostat. Neutrons interacting in LXe can easily escape outside the cryostat and travel inside the water tank. In order to enhance the neutron capture probability and consequently their detection efficiency, Gadolinium sulphate octahydrate ($Gd_2(SO_4)_3 \cdot 8(H_2O)$) is dissolved in water, for a relative mass concentration of 0.2%. Neutrons are thermalized within $\sim$20 cm from the cryostat and captured mostly by Gadolinium (due to the much larger cross section for

neutron capture compared to Hydrogen). An 8 MeV $\gamma$-ray cascade is emitted following the capture, which is finally converted into Cherenkov light in water.

To efficiently detect such neutron capture signals, an octagonal structure 3 m-high and 4 m-wide encloses the volume surrounding the TPC inside the water tank, defining the NV active region. The NV walls are made of expanded PTFE (ePTFE) reflective panels that provide >99% reflectivity to Cherenkov light above 280 nm wavelengths. In addition, 120 highly-sensitive and low-radioactivity 8" PMTs are distributed along the NV lateral walls with only the photocatode section protruding into the NV volume, in order to minimize the amount of radiogenic NV background events originating from the PMTs themselves. The outer cryostat vessel and the other objects inside the NV are cladded with ePTFE to maximize the light collection efficiency. The NV will allow for tag and reject neutron background events with an estimated efficiency of 87%.

### 5.2.5. Krypton and Radon Distillation

Natural Krypton, which contains the $\beta$-emitter isotope $^{85}$Kr, is removed from the LXe target by cryogenic distillation exploiting the 10.8 times larger vapor pressure of Kr compared to Xe at $-96$ °C. With a cryogenic distillation column, the more volatile Kr is collected at the top while Kr-depleted Xenon is extracted at the bottom. XENON1T set a requirement of <0.2 ppt [mol/mol] $^{nat}$Kr/Xe concentration, which implies a reduction of about 5 orders of magnitude, given a $^{nat}$Kr/Xe concentration of $\mathcal{O}(0.01)$ ppm in commercial high-purity Xe gas. To achieve this goal, a distillation column 2.8 m-tall was built following ultra-high vacuum standards. The reach of the system was demonstrated to be <48 ppq $^{nat}$Kr/Xe concentration [100]. The total height of the distillation system used in XENON1T is 5.5 m.

To allow for data acquisition with a fully operational dual-phase TPC while at the same time reducing the Kr concentration, XENON1T successfully established the online removal of Kr. After continuously operating in this mode for 70 days, with an initial measured $^{nat}$Kr/Xe concentration of 60 ppb, a minimum concentration of $0.36 \pm 0.06$ ppt [101] was measured with a gas chromatography system coupled to a mass spectrometer (rare gas mass spectrometer, RGMS [102]). This concentration is the lowest ever achieved in an LXe dark matter experiment.

The possibility for online removal of Radon as well was first demonstrated by installing a shortened version of the final cryogenic distillation column in reverse and lossless mode in XENON100, achieving a reduction factor of >27 [103]. The same principle was also demonstrated in XENON1T [104] by using the Kr distillation column in reverse mode to remove Radon. A dedicated Radon distillation column has been developed for online $^{222}$Rn distillation in XENONnT.

## 6. Physics Highlights from the XENON Experiments

Thanks to the most sensitive LXe TPCs ever built, the XENON collaboration has been able to probe unexplored regions of the WIMP parameter space, improving the previous best limits by more than 3 orders of magnitude thus far. The most stringent constraints on the SI WIMP-nucleon coupling obtained by the experiments of the XENON project are summarized in Figure 7, together with the projected sensitivity of XENONnT. Besides the standard SI and SD WIMP interactions, the XENON experiments constrained many other DM models and signatures: inelastic WIMP interactions [105–109], WIMP-pion coupling [110], EFT operators [111], leptophilic DM [112], axions and ALPs [113,114], bosonic Super-WIMPs [115], light mediators and dark photons [116], and annual modulation [117,118]. Moreover, XENON1T demonstrated that ultra-low background LXe detectors are also highly sensitive to ultra-rare processes not related to DM, such as double electron capture decays [78] and CE$\nu$NS [119], and that multi-ton LXe detectors can be competitive in the search for the neutrinoless double beta decay [63]. The latter channel will be one of the primary searches for the next-generation DARWIN detector [52], which will be ultimately limited by the irreducible CE$\nu$NS background.

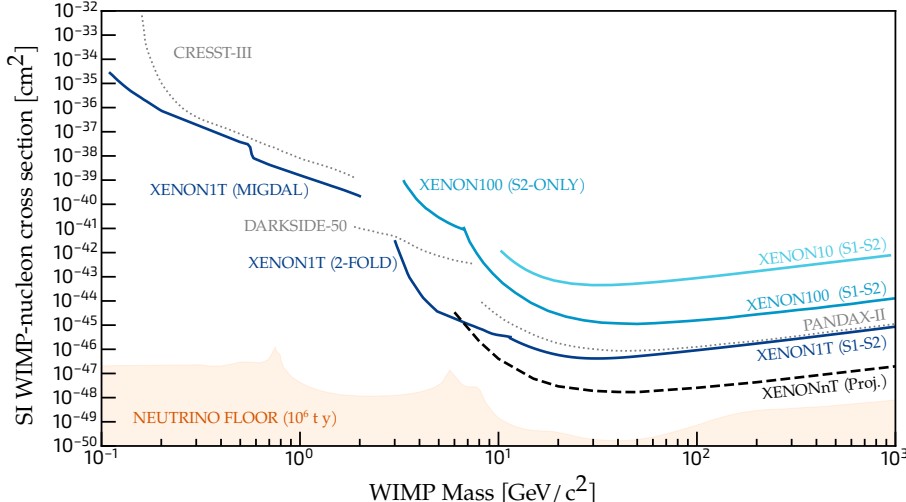

**Figure 7.** Best limits (90% confidence level) on the SI WIMP-nucleon cross section from the XENON experiments. At larger WIMP masses, standard DM searches based on both *S*1 and *S*2 signals are most sensitive. In order, XENON10 (2008) [120], cyan, XENON100 (2016) [121], azure, XENON1T (2018) [84], blue. The expected sensitivity of XENONnT [79] in the goal 20 t y exposure is shown as a black dashed line. All such analyses require a 3-fold PMT coincidence of the *S*1 signals. Low-mass WIMPs ($\lesssim 10$ GeV/$c^2$) are probed by releasing the 3-fold requirement in order to lower the experimental energy threshold. Both XENON100 and XENON1T conducted data analyses exploiting only the *S*2 signal. The *S*2-only XENON100 (2016) result [122], azure, is combined with the *S*1–*S*2 limit in a unique line, where the kink indicates the transition of the best limit between the two different analyses. The XENON1T's *S*2-only limit (2019) [116] (not shown) was improved by an *S*1–*S*2 search with 2-fold *S*1 coincidence requirement. The XENON1T 2-fold (2020) [119] curve is combined with the standard XENON1T's result (blue). The XENON1T Migdal (2019) limit [108], blue, was obtained with a standard *S*1–*S*2 analysis looking for enhanced low-energy signals due to the Migdal effect. XENON1T's results are currently the world-leading constraints in the WIMP mass range (0.1, 1000) GeV/$c^2$, but between 2 and 3 GeV/$c^2$ where DarkSide-50 (2018) [123], a liquid Argon TPC at LNGS, set the best limit. The dotted grey line shows the combination of the current most stringent limits excluding XENON1T. At lower WIMP masses, CRESST-III (2019) [124] (a cryogenic scintillating calorimeter at LNGS) achieved the highest sensitivity. In the ($\gtrsim 10$ GeV/$c^2$) WIMP mass regime, the exclusion limit by PANDAX-II (2017) [125] is very close to the result from LUX (2017 [126]), not shown for simplicity. Both detectors are LXe dual-phase TPCs as XENON experiments. The orange area shows the discovery limit of an idealized LXe experiment with the sole CE$\nu$NS background and $10^6$ t y exposure [127].

### 6.1. XENON10

XENON10 collected DM search data from October 2006 to February 2007 for a total of 58.4 live days. The blind data analysis was conducted selecting a 5.4 kg central fiducial LXe mass and restricted to the NR region of the parameter space (with ∼90% ER rejection and ∼45% NR acceptance). The statistical inference was based on the *maximum gap* method [128]. Previously unexplored parameter space was probed for both SI [120] and SD [129] WIMP-nucleon couplings, reaching the strongest constraint of $4.5 \times 10^{-44}$ cm$^2$ ($5 \times 10^{-39}$ cm$^2$) on the SI (SD to neutron-only) cross section of a 30 GeV/$c^2$ WIMP. The inelastic dark matter scenario was also tested with XENON10 [130] improving the previous limits and contributing to exclude the interpretation of the DAMA positive modulation signal [131] in terms of iDM (as such model was suggested to relieve the tension between the DAMA claim and the null results of other DD experiments).

*6.2. XENON100*

For the analysis of its first data [132], XENON100 introduced for the first time a profile likelihood (PL) approach to data analysis in the DD field [133]. Contrary to the maximum gap or other box-counting methods, the statistical inference based on PL allows for exploiting the full $S1$–$S2$ observable space, without hard cuts to reject backgrounds, and to naturally include systematic uncertainties. This requires the detailed modeling of the expected signal and backgrounds in the $S1$–$S2$ space, which was achieved by calibrating and modeling the detector response to both NRs [134] (with external $^{241}$AmBe neutron source) and ERs (with external $^{137}$Cs, $^{60}$Co, $^{232}$Th gamma sources and with internal Tritium beta source [135]). The full background model was then obtained from the convolution of the ER and NR response models with detailed MC simulations of external sources of ER [73] and NR [136] background, and data-driven studies of intrinsic Kr and Rn sources [137]. The XENON100 data analysis methods are detailed in Ref. [138].

Three science runs were collected by XENON100 over 4 years, between 2010 and 2014. Both the first (101 live days and 48 kg fiducial LXe mass, 2010) and the second run (225 live days and 34 kg fiducial mass, 2011–2012) set the strongest constrains on the SI [139,140] and SD WIMP-neutron [141] couplings at the time of publication. The final results from the three runs combined for a total of 477 live days set exclusion limits on the SI (SD) WIMP-nucleon (-neutron) cross section with a minimum of $1.1 \times 10^{-45}$ cm$^2$ ($2.0 \times 10^{-40}$ cm$^2$) at a WIMP mass of 50 GeV/c$^2$ [121].

By searching for signals in the ER channel, XENON100 improved the previous best limits on the axion and ALPs coupling to electrons [113]. Searches for annual modulation have been conducted thanks to the long-standing and stable data acquisition operations of XENON100 which reported a null result. The interpretation of the DAMA/LIBRA signal [142] in terms of DM coupling to electron via axial vector coupling was rejected with a significance of 5.7 $\sigma$. Moreover, other leptophilic DM models, such as mirror and luminous DM, have been excluded as explanation for that modulated signal with a significance > 3 $\sigma$ [112].

*6.3. XENON1T*

From November 2016 to February 2018, XENON1T collected dark matter search data in two science runs, called SR0 (32.1 live days) [101] and SR1 (246.7 live days), for a combined exposure of 1 t y [84].

The ER background in XENON1T was the lowest ever achieved in a DD experiment. In the low-energy region of interest for WIMP and axion searches, the measured ER rate was $76 \pm 2$ (t y keV)$^{-1}$ [114], in agreement with the expectation from MC simulations [60]. The good match between simulations and data held at higher energies as well (up to few MeV). The ER background characterization, its ultra-low level, and the good energy resolution (4% at 64 keV and $\sim 1\%$ in the MeV range [63]), allowed for the first ever observation of the DEC decay process [78], whose measured half-life is the longest ever measured (see also Section 5.1.1). Within the 1.3 t fiducial LXe mass used for DM searches, the ER background was dominated by intrinsic sources: the $\beta$-emitters $^{214}$Pb (75%) and $^{85}$Kr (11%). The NR background was dominated by radiogenic neutrons, with the highest contribution from TPC PMTs (50%). The contribution of neutrino interaction via CE$\nu$NS amounted to just 3%. The NR background expectation in the final 1 t y exposure was $1.4 \pm 0.7$ events, as estimated from MC simulations and constrained by the actual detection of multiple NR scatters.

The XENON1T response to low-energy ERs and NRs was characterized with internal $^{220}$Rn [143] and external neutron calibration sources (deuterium–deuterium plasma fusion neutron generator [144] and $^{241}$AmBe [62]). The model describing the light and charge signals' generation and propagation across the detector for both ER and NR interactions was simultaneously fitted to calibration data [64]. The derived separation between the ER and NR populations in the observable $S1$–$S2$ space corresponded to a 99.7% rejection power of ERs in a reference NR region defined between the NR median and the $-2\sigma$ quantile [64].

The inference on blinded data for SI and SD WIMP-nucleon interactions was conducted exploiting a likelihood function parametrized in four dimensions: $cS1$, $cS2$ (signal space), $r$, and $z$ (spatial coordinates) [145]. Since the data analysis method was established for XENON1T, confidence intervals on WIMP cross sections are calculated using the Feldman–Cousin construction [146] in order to get an exclusion limit or a two-sided interval (implying discovery) depending on the likelihood-based statistical test on data. Such method avoids the flip-flop problem due to the a priori decision on the reporting upper limit only at the cost of weaker exclusion power. A complete description of data analysis and statistical modeling techniques used in XENON1T is given in Refs. [64,145], respectively.

In the search for WIMPs, XENON1T observed no significant excess over the expected background in 279 live days and 1.3 t fiducial mass. World-leading exclusion limits on the SI WIMP-nucleon cross section were set for WIMP masses > 6 GeV/c$^2$, with a minimum of $4.1 \times 10^{-47}$ cm$^2$ [84]. XENON1T improved limits on the SI WIMP-nucleon cross section also for WIMPs as light as 3 GeV/c$^2$ with data analyses that exploited only the $S2$ signal [116] or lowered the standard requirement of coincident $S1$ signals from 3 to 2 PMTs [119]. Moreover, the search for SI WIMP interactions enhanced by the Migdal effect [108] excluded a new portion of the parameter space for very light WIMPs with mass in the range (0.1, 2) GeV/c$^2$. The most stringent limit to date was set on the SD WIMP-neutron cross section, reaching $6.3 \times 10^{-42}$ cm$^2$ for a WIMP mass of 30 GeV/c$^2$ [147].

Thanks to the unprecedented ultra-low ER background rate, XENON1T could also perform a highly sensitive search for axions in the low-energy ER channel. Based on the analysis of SR1 data in a 1 t fiducial mass (0.65 t y exposure), XENON1T reported on a significant excess of events (285) observed in the energy range (1, 7) keV over the expected background (232 ± 15) [114]. The excess was found to be compatible with solar axions inducing ERs via axioelectric effect with 3.4$\sigma$ significance. Three processes are expected to contribute to the solar axion flux: ABC interactions [148], a $^{57}$Fe nuclear transition [149] and Primakoff conversion of photons [150]. The observed XENON1T signal would imply a non-zero rate of either ABC or both $^{57}$Fe and Primakoff axions. However, the derived constraints on the related axions couplings are in strong tension with limits from indirect searches [151], even though the discrepancy could be relieved by underestimated systematical uncertainties in stellar cooling models. Another possible explanation for the XENON1T excess is the presence of Tritium, whose $\beta$-decay spectrum is compatible with the observed one at a 3.2$\sigma$ significance level. The required $6 \times 10^{-25}$ (mol/mol) concentration of Tritium in LXe is impossible to be measured independently, and it would be the first indication of an atmospheric background source in LXe TPCs. Therefore, Tritium could not be either confirmed or excluded at the present time. When a Tritium component is included as additional background, the significance of the solar axion interpretation lowers to 2.0$\sigma$. Similarly to axions, interactions of solar neutrinos with anomalous magnetic moments [152] fit the ER excess with 3.2$\sigma$ (0.9$\sigma$) in the absence (presence) of Tritium background, but constraints on the neutrino magnetic moment would be in tension with indirect searches [153] as well.

### 6.4. XENONnT

The XENONnT experiment is currently under commissioning and will start the first science run in the second half of 2021. The goal of the XENON Collaboration is to acquire DM search data with XENONnT for five years and collect a total exposure of 20 t y with the selection of a core 4 t LXe fiducial mass. XENONnT is expected to push the WIMP sensitivity by more than one order of magnitude compared to XENON1T, to the projected best limit on the SI cross section of $1.4 \times 10^{-48}$ cm$^2$ at 50 GeV/c$^2$ WIMP mass [79]. An analogous step forward is estimated on the SD WIMP coupling to neutrons, with a minimum sensitivity of $2.2 \times 10^{-43}$ cm$^2$.

Such improvement is the result of a lower background level, in addition to the larger LXe target and data acquisition time period. The predicted ER background rate

in XENONnT is $12.3 \pm 0.6$ (t y keV)$^{-1}$, about 6 times smaller than XENON1T. Such expectation assumes a target $^{222}$Rn contamination of 1 µBq/kg, which corresponds to a one order of magnitude suppression of the 11 µBq/kg level measured in XENON1T [114]. This is based on meticulous selection of materials with low Radon emanation, higher LXe volume-to-surface ratio in the TPC, and the novel online cryogenic Radon distillation column. The goal $^{nat}$Kr/Xe concentration is set at 0.1 ppt, well within the Krypton distillation system reach [100]. The radiogenic ER background is reduced by ∼30% with respect to XENON1T due to the larger LXe layer that shields the fiducial mass and the natural decay of $^{60}$Co (5.3 y half-life), in particular, in the detector components outside the TPC, as most of them are reused from XENON1T.

The neutron background is significantly suppressed with respect to XENON1T thanks to the larger LXe shield and multiple scatter probability (a factor ∼2 reduction) and the active Neutron Veto (a factor ∼7 suppression). In addition, 96% of the residual radiogenic NRs originates from the SS cryostat (36%), PMTs (33%) and PTFE components (26%). The irreducible CE$\nu$NS background is expected to be the dominant contribution affecting the sensitivity to low-mass WIMPs ($\lesssim 10$ GeV/c$^2$) due to solar neutrinos, while, for larger WIMP masses, ERs are the largest background source, with comparable rates expected from neutrons and CE$\nu$NS.

Thanks to the further reduced ER background level, XENONnT is expected to shed light on the XENON1T observed excess with its first science run. The solar axion hypothesis could be discriminated from the Tritium background with 5$\sigma$ significance with just a few months of data by the difference in the spectral shapes.

## 7. Conclusions

In this work, the dark matter DD approach is described with particular emphasis on the WIMP paradigma and in the context of Xenon-based detectors. The combination of a liquid Xenon target and the dual-phase TPC technology brings several advantages to the experimental sensitivity to DM and ultra-rare events in general. Xenon provides a high cross section to WIMP interactions and is an excellent scintillator. Xenon is also intrinsically radiopure, thus favoring the construction of detectors with ultra-low background levels. The simultaneous detection of scintillation and charge signals, enabled by dual-phase TPCs, provides very good energy resolution, high discrimination power between ERs and NRs, and the possibility to reconstruct the 3D position of interactions.

The XENON project is a world leader among DD experiments, thanks to the development of detectors with increasing size and decreasing background, from the prototype XENON10 (2005) to the current XENONnT (2020), all located at LNGS. The main features of the XENON detectors are described, highlighting the improvements introduced over time and focusing on the recent ton-scale XENON1T and XENONnT experiments. The relevant background sources are discussed along with the strategies adopted to achieve the lowest ever background level with XENON1T and to move further with XENONnT. The findings of all the concluded XENON experiments constrained previously unprobed parameter space of WIMP and other DM models. The main results obtained to date are presented and the expectations from XENONnT are discussed. The primary goals of the XENON Collaboration are to further improve upon the XENON1T world leading sensitivity to WIMPs by one order of magnitude and to shed light on the low-energy excess of ERs observed by XENON1T, in order to disentangle the Tritium background hypothesis from the solar axion (or anomalous neutrino magnetic moment) interpretation.

**Funding:** This research received no external funding.

**Institutional Review Board Statement:** Not applicable.

**Informed Consent Statement:** Not applicable.

**Data Availability Statement:** Not applicable.

**Conflicts of Interest:** The authors declare no conflict of interest..

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
