# Peer review of "The Xenon Road to Direct Detection of Dark Matter at LNGS: The XENON Project"

_universe, doi:10.3390/universe7080313_

Round 1

Reviewer 1 Report

The paper is a nice review of the techniques for direct detection of Dark Matter with liquid Xenon TPCs, from the detection principles to the description of the XENON detectors project, including a thorough discussion of the background contributions and mitigation, and the physics performances of XENON experiments.

I suggest the publication of the paper in MDPI Universe. I also suggest to implement minor corrections listed in the commented version of the manuscript pdf.

Author Response

Thank you for your review.

Please see the attachment for my responses and the latest version of the manuscript where changes are highlighted (changes in the bibliography are not highlighted, sorry, but additional references are visible in the text).

Reviewer 2 Report

Overall, I think this review of the experimental program of the XENON collaboration is well-suited to publication in this special issue of Universe. However, some revisions are in order before publication. Most importantly, I think the level of detail is not well-balanced between the different aspects of the XENON program this review covers.

Section 1: This introduction is very focused on the XENON program. For the purposes of a review, it would be useful to discuss in more detail how the XENON program fits into the general quest for dark matter, the WIMP direct detection program, and the landscape of liquid Xe based detectors. It would also be beneficial to discuss the evolution of the field in time, i.e. what was the state before the XENON10/100/1T/nT detectors and what are the next step(s) after XENONnT. What are other ideas to extend the dark matter reach to lower masses and/or smaller couplings? Finally, please add references to the introduction. In particular, the first few paragraphs should cite reviews of the evidence for dark matter, dark matter candidates, and the different experimental/observational approaches to searching for dark matter.

Section 2: There are quite a few technical issues in this section, see the detailed comments below. Please review this section carefully. More generally, this section turns relatively quickly into a liquid Xe direct detection focused discussion. It should either be stated that this discussion is applicable to the purposes of such an experiment only, or the section should be re-written more broadly.

Section 3: It might add some clarity to show a plot illustrating the ER/NR separation possible in dual phase Xenon TPCs due to the scintillation + ionization signal readout. Also, I do not understand the last paragraph of section 3.1, please clarify.

Section 4: Why is XENON1T described in much greater detail than XENON10/100/nT? It would be good to identify the most important details to discuss for each experiment, and use a consistent level of detail throughout section 4.

Section 5: This section contains a rather detailed discussion of the background sources as well as of the background mitigation strategies. Generally, sections 5.1 & 5.2 could be re-organized to fit a bit better together. More importantly, reading this five-page discussion of the backgrounds, it is rather difficult to keep track of the different backgrounds, their relative importance, and how well the different mitigation strategies work. In order to help orient the reader, it would be useful to include one (or multiple) figures or tables which illustrate the relative importance of the different backgrounds before and/or after the respective mitigation strategies and which provide a quick overview. Another plot that might be helpful here is an illustration of the ER/NR rejection mentioned in my comments for section 3. Furthermore, self-shielding is a particular advantage the large liquid noble gas detectors have over other direct detection experiments, and it is repeatedly discussed in section 5. It might be helpful to include a plot showing the survival probability of the different background components as a function of depth in liquid Xe or some other illustration of the self-shielding to explain this to the reader and to demonstrate why the physical size of the current generation of Xe detectors is important.

Section 6: This section discusses the most important physics results of the XENON program, interleaved with some discussion of the analysis techniques used to obtain these results. Given the level of detail with which other aspects of the XENON program are discussed in this review, I encourage the author to think about if it would not be better if the analysis techniques used by the collaboration are discussed in a little more detail in a separate section, and if the current section 6 would focus exclusively on the physics results.

Section 7: Similar to the introduction, in my opinion, it would serve the paper well to place the discussion of the XENON experiments in a broader context. Furthermore, the XENON collaborations has sufficiently impressive results, it is not necessary to make statements stretching the truth such as "Xenon provides the highest cross section to WIMP interactions" (l828). Other materials have higher spin-independent cross sections (any nucleus heavier than Xe) or larger spin-dependent cross sections (for example, F is much better suited to search for spin-dependent WIMP-proton scattering). The reason for the success of the Xe dual-phase program is due to a number of factors, including the large number of nucleons in Xe nuclei, but also many other reasons of which many are discussed in this review. Similarly, I find the various comments on the abstract and the introduction seemingly talking down other direct detection experiments by, e.g., stressing how "each detector of the XENON project obtained the best constraints" (l9) unnecessary. 

More detailed comments:

l41: The $\mu$eV--meV mass range quoted here is perhaps appropriate for QCD axion DM (even there it is debatable), but certainly this is far too restrictive for axion-like particle dark matter. (see arXiv:1201.5902 for a classical reference, although more recent works are available)

l47: "DM particles scattering off target atoms in deep underground laboratories" is a particular version of direct detection. In the intro, this should be discussed in a broader context.

l49: signals from dark matter annihilation are again a particular version of indirect detection. Other examples include looking for the products of dark matter decay (the prototypical example being sterile neutrinos) or looking for the products of the conversion of dark matter into Standard Model particles (the prototypical example being axion-photon conversion in astrophysical systems).

l81: The description of the different ways dark matter interactions deposit energy in a target material doesn't seem quite correct. First of all, everything discussed here is for WIMP-nucleus scattering. In that case, the interaction proceeds via dark matter scattering of a single nucleus. It is then that recoiling nucleus which in turn deposits its energy in the material via excitation of the other atomic nuclei, atomic ionization, or heat production.

l84: The split of which type of signal is observed with which type of detector is rather artificial. For example, here the author writes that the ionization signal is measured in Ge and Si semiconductor detectors, only to discuss a few lines later how most detectors actually use multiple readout channels in order to allow for ER/NR discrimination. 

l109: There seems to be a bit of a mixing here between the effects which come from the relative speed of the detector compared to the dark matter rest frame and the effects that come from the change in the direction of the detector relative to the dark matter wind. Also, the "wind" is responsible for both the annual and the diurnal modulation. Ref. [13] of the manuscript is the original paper proposing directional direct detection experiment. The diurnal modulation signal was first discussed by Collar and Avignone, Phys. Lett. B 275 (1992) 181-185.

l109: The original reference for the annual modulation signal is Drukier, Freese and Spergel Phys. Rev. D 33, 3495, not Ref. [12] of the manuscript.

l114: the track length quoted here as 1 mm is very detector dependent. Most importantly, there are crucial differences between solid-state/liquid and gaseous detectors that should be mentioned here.

l116: The reason that WIMP-nucleus scattering is typically considered as the search channel for WIMP DM has very little to do with the fact that dark matter "must be neutral". Note also that dark matter does not have to be strictly electromagnetically neutral, the simplest counterexample is millicharged dark matter.

l119: In Eq. (4) the scattering cross section is described in terms of the usual SI and SD interactions. While this is an oft-used parameterization, more general interactions are possible as the author discusses in Sec. 2.2.2.

l145: The statement that EFTs allow for a direct comparison between collider and direct detection searches for dark matter is not correct. In particular, the non-relativistic EFT discussed in this paragraph manifestly requires that the momentum transfer is small compared to 1) the mass of the nucleus, 2) the mass of the WIMP, 3) the mass of the mediator. Neither of these conditions is satisfied for dark matter production at a collider. Note that the EFT framework used in the analysis of the Run 1 LHC dark matter searches is very different from the non-relativistic EFT for the direct detection of dark matter. Furthermore, most LHC dark matter searches no longer make use of the previously used EFT framework, but have moved on to using simplified (or more complex) models since it was shown that even the EFT framework used for LHC analysis was not generally appropriate for the LHC.

l147: There are two different concepts that are mingled up in this paragraph. 1) inelastic dark matter typically refers to a model where the dark matter is inelastic, either because it is composite or because there is a second state in the dark sector that participates in the WIMP-nucleus scattering. 2) "ordinary" WIMPs can scatter inelastically off nuclei by exciting some internal degree of freedom of the atomic nucleus. This latter option has nothing to do with inelastic dark matter models.

l162: the statement that only nuclear recoils above ~1 keV are detectable is rather specific to the standard analysis of the liquid XENON detectors. For example, cryogenic bolometric detectors have demonstrated much smaller nuclear recoil energy thresholds of the order of 10 eV.

l162: there is a difference between leptophilic dark matter and dark matter that can scatter off electrons. Any typical WIMP can and will scatter off electrons, it is just that for the typical WIMP mass range and types of interactions the WIMP-nucleus scattering is experimentally easier to detect that WIMP-electron scattering. This situation is reversed at WIMP masses much below ~1 GeV. On the other hand, leptophilic DM typically refers to WIMP models which are designed to have suppressed interactions with nuclei while the interactions with electrons remain large. Such models were, for example, developed to explain the DAMA excess, and need not necessarily have dark matter masses in the sub-MeV mass range.

l190: please choose one unit of temperature and use that throughout the paper (Kelvin or Celsius)

l681: as far as I can see, this is the first time the DARWIN project is mentioned. Since it is the next step in the liquid Xe detector program, it might be useful to introduce DARWIN in the introduction in a discussion of how the XENON project sits in the wider experimental landscape.

l694: Similarly to the comment above, it might be good to introduce the DAMA excess and (the problems with) its interpretation as a dark matter signal much earlier in the paper, e.g., in the introduction.

l694: Ref. [113] is a somewhat strange choice here. There are newer results from the DAMA collaboration available. Please include also references to the attempts to explain the DAMA excess with inelastic DM models (and the latest updates of these analyses). Note that in contrast to what is stated here, XENON alone cannot excluded the inelastic dark matter explanation for DAMA, instead, CRESST data was required for this, see, e.g., arXiv:2102.08367.

l779: the statement "even though the discrepancy could be relieved by underestimated systematical uncertainties in stellar cooling models" seems misleading to me. As far as I know, the changes to stellar cooling rates from an axion-like particle explaining the XENON1T excess are much larger than what could plausibly be explained with systematic issues in modeling stellar evolution, see, for example, arXiv:2006.12487.

Author Response

(The authors gave the same response as above.)

Reviewer 3 Report

Dear Authors,

The Review is well written and will be interesting to students and physicists who are starting to work in the field of direct dark matter searches and interested in the XENON experiment.

I have just a few minor comments:

1) Add references in the introduction.

2) Equation (3): mn -> mN

3) Equation (4): Er -> ER

4) Check equation (5). A reference?

5) Line 145. Could you provide an example or a reference to "a direct comparison between DM searches at colliders and DD constraints" made within EFT framework which are not SI / SD?

6) Line 148. Electronic -> photon?

7) Line 198. Probably some formulas could be used for illustration.

8) Figure 1. Describe all shown electrodes.

9) Line 599. chians -> chains.

10) Line 772. 1,7 -> 1 - 7

  Thanks,

     a reviewer.

Author Response

(The authors gave the same response as above.)
